# Twelve-month prevalence of haemarthrosis and joint disease using the Haemophilia Joint Health score: evaluation of the UK National Haemophilia Database and Haemtrack patient reported data: an observational study

Richard A Wilkins [ID],[1,2] David Stephensen [ID],[3,4] Heidi Siddle,[1] Martin J Scott,[5,6] Hua Xiang,[7] Elizabeth Horn,[2] Ben Palmer,[7] Graham J Chapman,[8] Michael Richards,[2] Rebecca Walwyn,[9] Anthony Redmond[10]

For numbered affiliations see end of article.

**Correspondence to**
Dr Richard A Wilkins;
r.a.wilkins@leeds.ac.uk

## ABSTRACT

**Objectives** To report the 12-month prevalence of joint bleeds from the National Haemophilia Database (NHD) and Haemtrack, a patient-reported online treatment diary and concurrent joint disease status using the haemophilia joint health score (HJHS) at individual joint level, in children and adults with severe haemophilia A and B (HA/HB) without a current inhibitor.

**Design** A 2018 retrospective database study of NHD from which 2238 cases were identified, 463 patients had fully itemised HJHS of whom 273 were compliant in recording treatment using Haemtrack.

**Setting** England, Wales and Scotland, UK.

**Participants** Children (<18 years) and adults (≥18 years) with severe HA and HB (factor VIII/factor IX, <0.01 iu/mL) without a current inhibitor.

**Primary and secondary outcomes** Prevalence of joint haemarthrosis and concurrent joint health measured using the HJHS.

**Results** The median (IQR) age of children was 10 (6–13) and adults 40 (29–50) years. Haemarthrosis prevalence in HA/HB children was 33% and 47%, respectively, and 60% and 42%, respectively, in adults. The most common site of haemarthrosis in children was the knee in HA and ankle in HB. In adults, the incidence of haemarthrosis at the ankles and elbows was equal. The median total HJHS in HA/HB children was 0 and in adults with HA/HB, were 18 and 11, respectively. In adults with HA/HB, the median ankle HJHS of 4.0 was higher than the median HJHS of 1.0 for both the knee and elbow.

**Conclusion** Despite therapeutic advances, only two-thirds of children and one-third of adults were bleed-free, even in a UK cohort selected for high compliance with prophylaxis. The median HJHS of zero in children suggests joint health is relatively unaffected during childhood. In adults, bleed rates were highest in ankles and elbows, but the ankles led to substantially worse joint health scores.

### Strengths and limitations of this study

► This study reports the 12-month prevalence of haemarthrosis in children and adults with severe haemophilia without current inhibitors, and associated Haemophilia Joint Health Score (HJHS) as a measure of joint disease.

► Prevalence and site were collated retrospectively from Haemtrack and HJHS from the National Haemophilia Database.

► Only the most compliant of patients who were adherent to taking and reporting prophylaxis on a national electronic treatment diary Haemtrack with concurrent HJHS scores were included.

► Sample size was affected by methodology including those with electronic fully itemised HJHS and above 75% threshold of compliance.

► The design of this study does not allow examination of longitudinal joint bleed or joint health status.

## INTRODUCTION

Haemophilia is a rare X-linked recessive genetic disorder characterised by bleeding into soft tissue and joints.[1] The most common forms are haemophilia A (HA) and B (HB), affecting 1:10 000 and between 1:35 000 and 1:50 000, respectively. The disease is further characterised by the levels of factor VIII (FVIII) and factor VIX (FVIX), with the most severely affected having less than 1% (<0.01 IU/mL) circulating clotting factor (severe haemophilia).[2] Musculoskeletal bleeding is the most common haemorrhagic manifestation, with 90% of bleeds occurring in muscles or joints.[1] The presence of blood products

within the joint space and the process of removal leads to synovial hypertrophy, haemosiderin deposition and eventually arthropathic joint changes.[3] Over time, repeated haemarthrosis results in chronic synovitis, changes in cartilage and bone composition and progressive chronic haemarthropathy.[4 5]

Infusion of replacement coagulation factor concentrates (CFC) is prescribed with the aim of elevating circulating factor to a level that halts spontaneous and traumatic bleeding.[1] CFC treatment is not without complication. The development of antifactor antibodies or 'inhibitors' in some people produces an immune response to CFC infusion that significantly reduces the effectiveness of CFC treatment. Development of inhibitors increase the risk of bleeding, joint damage and requirement for factor treatment bypassing agents.[6] Ultimately, the aim of modern treatment of haemophilia is prevention of joint bleeds with a target of achieving zero bleeds whenever possible. Prevention of haemarthrosis in all age groups is important and in particular in children, where musculoskeletal immaturity exposes joints to greater risk of damage in later life. Multiple studies have shown that early initiation of CFC prophylaxis in children delays joint damage and reduces joint disease.[7–10] In adults, multijoint haemarthropathy remains a common feature of the disease, but even prophylaxis started in adulthood decreases bleeding, improves pain and improves health-related quality of life.[11] Therefore, in children and adults prophylaxis is considered the standard of care for all patients.[11 12] Traditionally, prophylactic treatment in severe haemophilia aims to maintain FVIII or Factor IX (FIX) at a trough level >0.01 iu/mL. It is apparent that many patients experience spontaneous as well as traumatic bleeds, despite achieving trough factor levels >0.01 IU/mL. Several approaches have been adopted or are being investigated with the aim of attaining complete bleed avoidance, including more individualised treatment with standard half-life products, the use of coagulation factors with extended half-lives, and innovative non-factor treatments.[12–15]

Recent evaluation of real-world treatment regimes in severe and moderate haemophilia in the UK and Europe, has shown that despite adequate CFC availability, treatment is still suboptimal. In 2015, data from the UK National Haemophilia Database (NHD) reported median (IQR) annualised bleed rates (ABR)/annualised joint bleed rates (AJBR) in children (0–11 years) and adolescents (12–18 years) of 1.0 (0.0–0.5)/0.0 (0.0–1.0) and 2.0 (0.0–7.0)/1.0 (0.0–3.0), respectively. ABR in adults with severe haemophilia A on prophylaxis were 2.0 (IQR 0.0–7.0) and AJBR was 1.0 (IQR 0.0–4.0) with only 29% bleed free and 34% joint bleed free.[16] Similarly, reported European (Belgium, France, Germany, Italy, Spain, Sweden and UK) data shows median AJBR of 1.0–4.0.[16 17] However, data on bleeding frequency and severity of haemarthropathy at an individual joint level are lacking.

The main sites of haemarthrosis are the elbows, ankles and the knees, with the shoulders, wrists and hips less commonly affected and data for these sites not collated by the NHD. The Haemophilia Joint Health Score (HJHS) is a standardised clinical assessment tool developed to assess upper and lower limb joint health status. The clinical assessments undertaken by specialist physiotherapists at 6–12 months intervals include measurement of swelling, alignment, range of motion, and muscle atrophy, and forms part of the UKHCDO haemophilia management guidelines.[18 19] The HJHS is the most widely used score of joint health in haemophilia and has shown good to moderate correlations with radiological scores of joint disease using the Pettersson score.[18] However, haemarthrosis is not reported by the HJHS, and therefore, incidence of haemarthrosis and joint disease at an individual joint level are unknown.[20]

Those deemed most compliant with prophylaxis are less likely to experience repeated incidents of haemarthrosis and therefore less likely to have established joint disease when compared with those who do not adhere to treatment. This may be a smaller proportion than those who do not adhere to treatment but these cases are important in gauging the efficacy of current treatments.[11 19 20] Understanding prevalence and joint disease in the most compliant of patients may provide direction for future research of patient compliance and management of joint disease, including non-pharmacological interventions and intra-articular therapies commonly used in the management of MSK conditions.

## Objective

The primary objective of this study is to determine the prevalence and incidence of joint bleeding and joint disease using the HJHS at an individual joint level in children and adults with severe HA and HB without a current inhibitor.

## METHODS

The study has been reported in accordance with the UKHCDO NHD guidelines and regulations.

Data on bleed prevalence and site were collated retrospectively from the Haemtrack patient therapy recording system and the clinical HJHS from the NHD. Haemtrack is a UK national online treatment diary in which individual patients regularly report details of treatments with CFC.[20 21] Details of home delivery of CFC treatment to patients is recorded by the corresponding haemophilia treatment centre (HC) and then uploaded to the NHD. When CFC is administered by the patient that individual treatment is then recorded on Haemtrack, including the reason for each treatment such as prophylaxis or bleed treatment and the site of each bleed. Data recorded in Haemtrack are then integrated with NHD.[20] The 2018–2019 UKHCDO report indicated median compliance at haemophilia comprehensive care centres (CCC) and HC of 90% and 93%, respectively, with the NHD definition of compliance recorded use of ≥75% of received factor

concentrate.[22] The HJHS V.2.1 is collated as six individual joint scores (0–20) and compiled with a global gait score (0–4) to a total score (0–124). A higher HJHS score represents worse joint health.

Participants were children (<18 years old) and adults (≥18 years old) with severe HA and HB (FVIII or FIX <0.01 IU/mL) without a current inhibitor, who had been issued with CFC in the UK between 1 January 2018 and 31 December 2018. Regular prophylaxis was defined for those using standard half-life (SHL) prophylaxis as ≥2 infusions per week for HA, and ≥1 infusions/week for HB for >45 weeks/year; for patients using extended half-life (EHL) products, ≥1 infusions/week for haemophilia A, and more than once every 2 weeks for haemophilia B for >45 weeks/year. Low-dose prophylaxis is not prescribed in the UK, therefore, prophylaxis was assumed as above 25 IU/kg to maintain a trough level above 0.02 IU/mL.[23] Those included in the analysis were Haemtrack compliant (defined as recorded use of ≥75% of received factor concentrate) with a corresponding electronically recorded HJHS V.2.1.

The joint bleed prevalence (%) for paediatric and adult patients and AJBR and HJHS were collated from Haemtrack and NHD. AJBRs were reported by patients through the Haemtrack and recorded over the 12-month study period (1 January 2018 to 31 December 2018). Adequate primary and secondary prophylaxis and adherence to treatment are known to reduce bleed rates and reduce the burden of joint disease.[11 19] Therefore, only data from the most compliant patients (≥75% received factor concentrate vs recorded in Haemtrack) were reported as per the NHD standard operating procedure for data analysis and reporting. Joint bleed prevalence, AJBR and HJHS are reported for all joints (total) and in each individual joint. Data are summarised using means and SD or medians and IQRs (IQR 25–75 percentiles).

### Patient and public involvement
Patients from the Leeds Haemophilia Comprehensive Care Centre, Leeds, UK and The NIHR Leeds Biomedical Research Centre, Leeds, UK were involved in the original design of the author's clinical doctoral research fellowship and this original article.

## RESULTS
During 2018, 2238 individuals with severe HA (n=1889) and B (n=349) without a current inhibitor were registered with the NHD and 1396 were registered with Haemtrack. Electronically recorded fully itemised HJHS data were available for 463 patients with contemporaneous Haemtrack available for 273 individuals of whom 86.8% (n=237) had HA and 13.2% (n=36) HB. Participant age and treatment characteristics are presented in table 1.

### Joint bleed prevalence and ABR
Joint bleed prevalence (%) and individual joint prevalence, and total AJBR are presented in table 2. Bleed data are categorised by age, haemophilia type (A and B) and the most commonly effect joints (left and right) of the elbows, knees and ankles. Joint bleed prevalence in children with HA (32.5%) and HB (47.1%) reported at least one incidence of joint bleeding over the 12-month study period. Adults with HA (59.9%) and B (42.1%) reported at least one bleed over the same time period. Median AJBR at individual joints for children and adults were 0.0 (0.0;0.0) with the exception of the left ankle in children with HB (0.0; 1.0). Mean AJBR for adults and children at the ankles, knees and elbows are presented in figure 1.

### Haemophilia Joint Health Score
HJHS categorised by age, haemophilia type and joint are presented in table 3. Median (IQR) of HJHS in children were 0.0 (0.0; 0.0) in both HA and HB. In adults the total HJHS were higher than in children; the total HJHS is higher in HA than HB. At an individual joint level median (IQR) ankle HJHS of 4.0 (0.0; 8.0) were higher than for the knee 2.9 (4.1)/ 1.00 (0.0; 5.0) and elbow 3.3 (4.1)/ 1.0 (0.0; 7.0).

## DISCUSSION
In this study, we report the current prevalence of haemarthrosis in children and adults with severe haemophilia without current inhibitors, and associated HJHS as a measure of joint disease. The study was conducted retrospectively, using data from 2018 in a national database. In a national cohort of 2338 individuals, 463 patients had electronically recorded fully itemised HJHS, with the sample size further reduced to 273 patients who met the fully Haemtrack compliant criteria. During the

| Table 1 | Participant characteristics | | | | |
|---|---|---|---|---|---|
| | **Haemophilia A** | | **Haemophilia B** | | |
| **Patient characteristics** | **Age <18 (n=80)** | **Age ≥18 (n=157)** | **Age <18 (n=17)** | **Age ≥18 (n=19)** | |
| Age (median, IQR) | 10 (7–13) | 40 (29–50) | 12 (7–14) | 45 (25–48) | |
| SHL | 67% (n=54) | 77% (n=121) | 18% (n=3) | 32% (n=6) | |
| EHL | 29% (n=23) | 23% (n=36) | 70% (n=12) | 42% (n=8) | |
| SHL-EHL | 4% (n=3) | 0% | 12% (n=2) | 26% (n=5) | |

EHL, extended half-life product; IQR, Interquartile range; SHL, standard half-life product.

**Table 2** Annual joint bleed prevalence and AJBR of children and adults

| Annual joint bleed prevalence | | | Haemophilia A | | Haemophilia B | |
|---|---|---|---|---|---|---|
| | | | Age <18 (n=80) | Age ≥18 (n=157) | Age <18 (n=17) | Age ≥18 (n=19) |
| AJBR | All joints | Median (IQR) | 0.0 (0.0–1.0) | 1.0 (0.0–4.4) | 0.0 (0.0–2.0) | 0.0 (0.0–3.5) |
| Joint bleed prevalence | All joints | n (%) | 26 (32.5) | 94 (59.9) | 8 (47.1) | 8 (42.1) |
| | Right ankle | n (%) | 2 (2.5) | 27 (17.2) | 1 (5.9) | 2 (10.5) |
| | Left ankle | n (%) | 5 (6.3) | 35 (22.3) | 5 (29.4) | 2 (10.5) |
| | Right knee | n (%) | 13 (16.3) | 27 (17.2) | 1 (5.9) | 2 (10.5) |
| | Left knee | n (%) | 7 (8.8) | 24 (15.3) | 1 (5.9) | 2 (10.5) |
| | Right elbow | n (%) | 6 (8.0) | 29 (18.5) | 1 (5.9) | 3 (15.8) |
| | Left elbow | n (%) | 4 (5.0) | 35 (22.3) | 1 (5.9) | 2 (10.5) |

Joint bleed prevalence (%): numerator=number of patients who had bleeds, denominator=total cohort number.
AJBR, annualised joint bleed rates; IQR, Interquartile range.

data collection period, 62% of the national cohort used Haemtrack, 20% of whom fulfilled compliance criteria set by the NHD, permitting analysis of haemarthrosis and joint health status of a representative sample of UK with severe haemophilia without inhibitors. The sample size while small is focused only the most compliant of patients and provides insight to the current compliance rates and reporting of joint diseases to the NHD. The results presented in this paper represent the likely best case scenario for the most compliant cases and this further highlights the 80% of patients who fail to either record or comply with treatment and raises questions as to the real compliance and adherence to treatment, as well as the concurrent joint disease in patients who do not meet the 75% NHD inclusion threshold.

In children with severe haemophilia, average AJBR were low across haemophilia types. One in three children did however experience a joint bleed during the 12-month data collection period. The majority of those included would have typically been provided prophylaxis from an early age and continue to adhere to a prophylaxis regime, but 30% of children still experienced haemarthrosis during the 12 month data collection

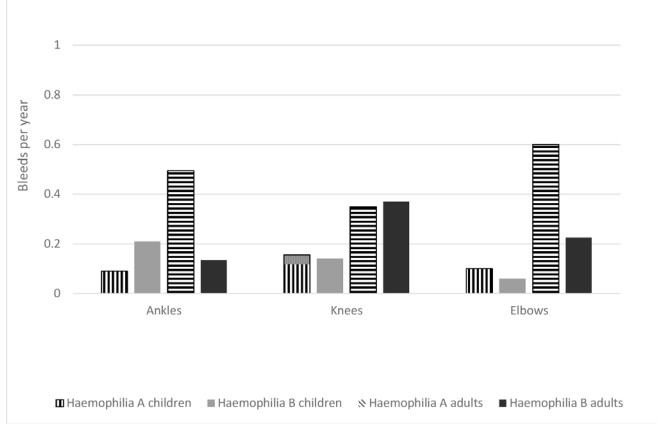

**Figure 1** Combined annual joint bleed rate for children (vertical and horizontal black columns) and adults (solid grey and black columns) with severe haemophilia A and B.

period. HJHS itemised by joint were very low in children (table 3) suggesting either minimal joint disease or that the HJHS might not be sensitive to early joint changes following haemarthrosis. Reliability of the HJHS has been explored in children and young adults and is reported to be sensitive to early joint changes,[24 25] although individual joint HJHS of less than three at the knee and ankle are less able to identify pathological joint change when compared with magnetic resonance and ultrasound imaging.[18] Similarly in children, correlations between the HJHS and the Haemophilia Early Arthropathy Detection with UltraSound (HEAD-US) have shown good correlations in the identification of joint pathology at the elbows and knees, however, at the ankles significant difference are reported between HJHS and HEAD-US scores with under-reporting of ankle joint pathology in both instances.[26] Therefore, a combined approach to joint health assessment may identify pathology especially at the ankle joint prior to the progression to haemarthropathy. Canine, mouse and human in vitro models have demonstrated chondrocyte apoptosis and reduced proteoglycan synthesis affecting cartilage matrix turnover within 48–96 hours of an induced joint bleed, suggesting a single joint bleed may have detrimental effects on joint cartilage.[27–29] Formally reported bleed rates in the NHD are relatively low, however, microbleeding (subclinical bleeding not clinically detectable, or experienced by the patient) is an emerging theme in haemophilia. Episodes of subclinical bleeding may contribute to the deterioration of joint health despite no clinically detectable signs of a joint bleed, therefore, point-of-care ultrasound tools such as the HEAD-US may provide early evidence of joint disease.[3]

In the adult population, AJBRs were higher than those reported in children, with mean (SD) AJBR of 3.9 (7.0) and median (IQR) 1.0 (0.0–4.4) in HA and 2.0 (3.6) and 0.0 (0.0–3.5) in HB, respectively. The 12-month prevalence was also higher, with 60% and 41% of adults with HA and HB, respectively, experiencing at least one bleed over the period. HJHS scores at the ankle joint were similar to

**Table 3** Haemophilia joint health scores for children and adults

| Haemophilia joint health scores | Haemophilia A | | Haemophilia B | |
|---|---|---|---|---|
| Median (IQR) | Age <18 (n=80) | Age ≥18 (n=157) | Age <18 (n=17) | Age ≥18 (n=19) |
| All joints | 0.0 (0.0–0.0) | 18.0 (6.0–31.0) | 0.0 (0.0–0.0) | 11.0 (5.0–24.0) |
| Right ankle | 0.0 (0.0–0.0) | 4.0 (0.0–8.0) | 0.0 (0.0–0.0) | 2.0 (0.0–7.0) |
| Left ankle | 0.0 (0.0–0.0) | 4.0 (0.0–8.0) | 0.0 (0.0–0.0) | 4.0 (1.0–8.0) |
| Right knee | 0.0 (0.0–0.0) | 1.0 (0.0–4.0) | 0.0 (0.0–0.0) | 0.0 (0.0–1.0) |
| Left knee | 0.0 (0.0–0.0) | 1.00 (0.0–5.0) | 0.0 (0.0–0.0) | 0.0 (0.0–2.0) |
| Right elbow | 0.0 (0.0–0.0) | 1.0 (0.0–7.0) | 0.0 (0.0–0.0) | 0.0 (0.0–1.0) |
| Left elbow | 0.0 (0.0–0.0) | 1.0 (0.0–6.0) | 0.0 (0.0–0.0) | 0.0 (0.0–1.0) |

HJHS: Global Gait score not included.

IQR, Interquartile range.

the elbows, with knees slightly less affected. Interestingly the median scores at both the knee and elbow were lower than that of the ankle, suggesting that there is worse ankle joint health overall when compared with other joints. Ankle joint changes are driven by the mechanical demand on the ankle and forces exerted on the joint during activities of daily living, in combination with structural and functional changes often seen in adolescents and adults with severe haemophilia.[30 31] Our data suggest that very early signs of joint disease might not be detected by the HJHS; rather it measures the cumulative effect of haemarthropathy, not detectable until later years.

AJBR in this study are slightly lower (table 2) than those reported in the UK THUNDER study conducted 3 years earlier using the same NHD database.[11] Scott *et al* reported a median AJBR of 0.0 in children (0–11 years), 1.0 in adolescents (12–18 years) and 3.0 in adults aged 19 and above. Our prevalence data (table 1) for both children and adults indicate a slight decrease AJBR since the Scott *et al* study.[16] In terms of the treatment profile of those included in our study, about one-quarter were now using an EHL product and 96% of those sampled are receiving and are compliant with treatment. In addition, Scott *et al* did not include those patients with HB who are reported to have better joint health and less frequent joint bleeds.[32] A longitudinal evaluation of tailored frequency-escalated prophylaxis in a Canadian cohort of children aged 1.0–2.5 years (n=36) followed up over 10.2 years (IQR 8.5–13.6) reported median index annual haemarthrosis rates of 0.95 (0.44–1.35) which is similar to our own results. Prophylaxis treatment in Canada was driven by bleed incidence and escalated accordingly, so their treatment was more targeted and reactive.[33] The Canadian study shows that avoidance of all joint bleeding is unlikely to be possible, and in our own cohort the mean (SD) AJBR of 0.81 (1.68) and 1.00 (1.18) in HA and HB children, respectively, indicate that bleeding is occurring in some children even when compliant with prophylaxis. In a Dutch study of haemophiliac adults (n=62) over a period of 5–10 years with a low median AJBR (IQR) 0.0 (0.0–2.0) there was still a worsening of joint health, with

a HJHS increase of more than four points over the study period in 37.1% of patients, and with the ankle joints most often affected (30.6%).[34] Those adults sampled in this study still had up to four joint bleeds over a 12-month period, with 60% of all adults reporting a minimum of one joint bleed. Forty per cent of individuals sampled reported no bleeds and were well controlled, but for the remaining 60% it is unclear why joint bleeding occurred. Understanding why the 60% in this cohort reported haemarthrosis may lead to better targeted and individualised treatment and identification of other contributing factors such as lifestyle and altered, combined and individual joint biomechanics of the upper and lower limbs.

A limitation of this study is the low proportion of patients registered on the UK database that had full Haemtrack and itemised HJHS data recorded at the time of data collection. The NHD does not report bleed level data on patients who do not use Haemtrack owing to the difficulty in collecting data from paper diaries and established links at each HC through the NHD Haemophilia Centre Information System, limiting analysis to Haemtrack compliant users.[20] Bias may have been introduced by the study design through the inability to include those not recording treatment in Haemtrack and those for whom HJHS examinations were not reported or itemised by joint to the NHD. Although this is the largest reported dataset of HJHS, the lack of linkage between elements of the data limits its wider utility. As electronic reporting of HJHS to the NHD becomes more routine and the dataset expands, we will be able link HJHS and joint health to rates of haemarthrosis. Haemtrack data compliance is defined as ≥75% of home delivery treatment received being recorded as used by the patient and so those who met the inclusion criteria are regarded as 'good reporters' and deemed likely to be compliant with treatment.[20] The current bleeding and joint disease profiles of those who receive and record treatment, but fall below the 75% treatment adherence criteria is unknown.

Access to individual treatment dose and trough levels were not available from the database and is acknowledged as a limitation of this study. Reporting of these data relies

on HCs uploading real time data, including trough levels and up to date measurements of weight but requires access to patient's data and requires better reporting methods to be achievable. Understanding joint haemarthrosis in this subset of patients may provide further insight into the real-world prevalence of haemarthrosis. This study focusses, for databasing reasons, on the most compliant cases and therefore those within the broader haemophilia population likely to be suffering the fewest consequences. It might be reasonable to expect that over the 12-month study period, comparable patients who do not report or full comply with treatment may have had higher bleed rates. Consequently it would also be expected that joint health may also be worse or deteriorating at a faster rate. Compliance is important because it represents a gap between the availability of best treatment and impact of treatment on the consequences. Less compliant patients may require different behavioural or system-based approaches to encourage compliance and better reporting and monitoring.

As expected due to the lower prevalence, the sample of HB patients in this analysis is smaller than the HA cohort, and therefore, differences in joint bleed prevalence and HJHS between patients with HA and HB should be interpreted with caution. Those with HB may present with a milder bleeding phenotype than that of HA regardless of severity or treatment.[32 35 36] In addition people with HB may display less severe levels of haemarthropathy, with differences in the specific pathophysiological mechanisms of joint disease underlined by different rates of joint deterioration and severity.[37] Direct comparison between disease types is limited and therefore further research is needed to explore whether the lower bleed rates and better joint health in people with HB suggested in this study can be confirmed.

History of spontaneous and traumatic bleeding could not be separated, owing to data reporting methods within Haemtrack. While prophylaxis protects against spontaneous bleeding there is still a proportion of these treatment compliant adults reporting up to four joint bleeds in the 12-month study period. Haemarthrosis may occur as individual joint events, but our data highlights the burden on overall joint disease. A history of developing inhibitors and a history of on-demand treatment now using secondary prophylaxis may predispose patients to higher levels of joint disease and greater risk of subsequent haemarthrosis.[11] Further research is required therefore to understand the bleeding profile and burden of disease in adults with established joint disease and previous inhibitor status.

A further limitation is between-centre variability in HJHS assessment.[38] HJHS data from different HCs may be subject to intercentre scoring variability, although workshops have been conducted in the UK to decrease intercentre variability in HJHS scoring. Furthermore, we are unable to confirm the influence of other factors such as the presence of comorbid musculoskeletal conditions on HJHS data. UKHCDO NHD data was also requested from those with moderate disease but there was insufficient data to include in the analysis. Future comparison by disease severity (severe and moderate) may provide further insight of those most at risk of haemarthropathy.

## Clinical implication and conclusion

In a UK cohort of Haemtrack compliant patients with severe haemophilia and without a current inhibitor, only 70% of children and 30% of adults remained haemarthrosis free during 2018. Haemarthrosis was most likely to be reported in the knee joint in children with HA, the ankle joint in children with HB, the elbow and ankle joint in adults with HA and the elbow joint in adults with HB. Overall higher HJHS were reported for the ankle joint compared with the knee and elbow, suggesting that the ankle joint is the most severely compromised joint in people with haemophilia.

Investigation of impact on function and potential interventions that lessen the burden of disease are warranted. Future clinical studies would also benefit from understanding the bleeding profiles of those who do not meet compliance criteria for Haemtrack or other database-linked bleed data to obtain the true prevalence of haemarthrosis and joint disease.

**Author affiliations**

[1]Leeds Institute of Rheumatic and Musculoskeletal Medicine, University of Leeds, Leeds, UK

[2]Leeds Haemophilia Comprehensive Care Centre, Leeds Teaching Hospitals NHS Trust, Leeds, UK

[3]Haemophilia Centre, East Kent Hospitals University NHS Foundation Trust, Canterbury, UK

[4]Haemophilia Centre, Barts Health NHS Trust, London, UK

[5]University Department of Clinical Haematology, Manchester Royal Infirmary, Manchester, UK

[6]Institute of Cancer Sciences, Faculty of Biology, Medicine and Health, The University of Manchester, Manchester, UK

[7]National Haemophilia Database, United Kingdom Haemophilia Centre Doctors' Organisation (UKHCDO), Manchester, UK

[8]School of Sport and Health Sciences, University of Central Lancashire, Preston, UK

[9]Clinical Trials Research Unit, Leeds Institute of Clinical Trials Research, University of Leeds, Leeds, UK

[10]NIHR Leeds Biomedical Research Centre, Leeds Teaching Hospitals NHS Trust, Leeds, UK

**Acknowledgements** The authors would like to thank the centre directors, physiotherapists and staff of all UK haemophilia centres for their hard work in collecting data and responding to data queries. We would also like to thank the members of the UKHCDO Data Analysis Group for advising on data interpretation; Prof Charles Hay, Prof Peter Collins, Dr Elizabeth Chalmers, Dr Ryan Cheal, Prof Pratima Chowdary, Simon Fletcher, Dr Georgina Hall, Dr Dan Hart, Dr Ri Liesner, Andrew McNally, Ben Palmer, Paul Sartain, Dr Martin Scott, Dr Susie Shapiro, David Stephensen, Dr Hua Xiang. The staff at the National Haemophilia Database are also thanked for their contribution to data collection.

**Contributors** The study was conceived by RAW, AR, GJC, RW and HS. Analysis was undertaken by members of staff at the NHD (HX and BP). The manuscript was written by RAW and DS. Subsequent drafts were edited and approved by RAW, DS, MJS, AR, GJC, MR, EH, HS and RW. RAW is responsible for the overall content of this article.

**Funding** RAW is funded by a National Institute for Health Research (NIHR) Clinical Doctoral Research Fellowship (ICA-CDRF-2015-01-0120); AR is in part supported through the NIHR Leeds Biomedical Research Centre, Leeds, UK.

**Disclaimer** This paper presents independent research funded by the National Institute for Health Research (NIHR). The views expressed are those of the authors and not necessarily those of the NHS, the NIHR or the Department of Health.

**Competing interests** RAW has received registration fees and support for travel from Roche. DS has received research funding from Sobi, CSL and Roche; consultancy and speakers fees from Sobi and Takeda. EH has received speaker fees from Roche, sponsorship for travel from Sobi. MJS has received research funding from Bayer; consultancy and speakers fees from Sobi and Roche; registration fees and support for travel from Sobi, Pfizer and CSL. HS is an HEE/NIHR Senior Clinical lecturer and has received funding from NIHR who also funded this research. AR is an NIHR Senior Investigator and has received funding from NIHR who also funded this research.

**Patient and public involvement** Patients and/or the public were involved in the design, or conduct, or reporting, or dissemination plans of this research. Refer to the Methods section for further details.

**Patient consent for publication** Not applicable.

**Ethics approval** Ethical approval was obtained to allow access the National Haemophilia Database, anonymised data. This study was approved by London Queen Square Research Ethics Committee (16/LO/2251) and NHS Health research Authority (IRAS ID 206141).

**Provenance and peer review** Not commissioned; externally peer reviewed.

**Data availability statement** Data are available on reasonable request. The data that support the findings of this study are available from The National Haemophilia Database (NHD) but restrictions apply to the availability of these data, which were used under license for the current study, and so are not publicly available. Data are however available from the corresponding author on reasonable request and with permission of the NHD.

**ORCID iDs**
Richard A Wilkins http://orcid.org/0000-0003-1885-5472
David Stephensen http://orcid.org/0000-0002-6175-3343

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
