## [Reviewer comments · BMJ Open]

ARTICLE DETAILS

TITLE (PROVISIONAL)	Twelve-month prevalence of haemarthrosis and joint disease using the haemophilia joint health score; evaluation of the UK National Haemophilia Database and Haemtrack patient reported data: an observational study
AUTHORS	Wilkins, Richard; Stephensen, David; Siddle, Heidi; Scott, Martin; Xiang, Hua; Horn, Elizabeth; Palmer, Ben; Chapman, Graham; Richards, Michael; Walwyn, Rebecca; Redmond, Anthony

VERSION 1 – REVIEW

REVIEWER	Can Ugur, Mehmet Izmir Bozyaka Training and Research Hospital, Hematology
REVIEW RETURNED	09-Jun-2021

GENERAL COMMENTS	I suggest re-evaluating it from a statistical point of view. Although the prevalence of hemarthrosis in children is 33%, the HJHS score of 0.0 is a contradictory situation. Also, how could the HJSH score be calculated when the data was recorded with an electronic diary? More information on this subject should be provided by the authors. I suggest re-evaluation after revisions.
--

REVIEWER	Kuijlaars, Isolde A.R. Utrecht University
REVIEW RETURNED	16-Jun-2021

GENERAL COMMENTS	I would like to thank the authors for this paper. This paper is an observational study on joint bleeding and joint health in patients with haemophilia, with a large number of data in terms of a rare disease like haemophilia. Although the current study described joint bleeds and joint status on an individual joint level, the final conclusion of this paper is not a very new insight for haemophilia care-givers. It would be very interesting to see some data and/or some more ideas about the excluded patients who are not compliant to Haemtrack. In addition, I am interested in the idea of the authors about annual HJHS assessment in children as the scores were very low. Abstract: - Please explain 'Haemtrack compliant' in the abstract. Introduction: - Please clarify in more detail what this study adds to current evidence on joint bleeds and joint health and why this is interesting for the haemophilia field. - health related QoL = health related quality of life - page 6, lines 56/58: factor vs. Faktor
---

	Introduction/methods: - Please check the first use of the abbreviation HJHS Methods: - I would suggest to describe the HJHS in more detail in de methods section Results: - Table 2 is a large table. I would recommend to include more details in the paragraph about this topic to guide the reader. - Could it be of interest to test the differences in joint bleeds between subgroups? - HJHS: I would recommend to test the differences in HJHS scores between children vs. adults, and between joints. - Table 2 and 3: please choose for the mean or the median. Discussion: - Page 14, line 18: HT? - Please add a comparison of the results to other Western populations. - What do the authors expect about the patients which are included in this study vs patients not included because they were Haemtrack non compliant?
--	--

VERSION 1 – AUTHOR RESPONSE

Dr. Mehmet Can Ugur, Izmir Bozyaka Training and Research Hospital	
I suggest re-evaluating it from a statistical point of view. Although the prevalence of hemarthrosis in children is 33%, the HJHS score of 0.0 is a contradictory situation.	Haemarthrosis prevalence represents that 33% of patients sampled had at least one incident of joint bleeding during the 12 month period. The HJHS score, a measure of joint health, represents the clinical measure of joint disease which may or may not arise after a bleed. The HJHS of zero is not contradictory it just suggests that there are no significant joint changes occurring in childhood, or that they are not clinically detectable
Also, how could the HJSH score be calculated when the data was recorded with an electronic diary? More information on this subject should be provided by the authors.	The HJHS score is not recorded in the online diary, it is a clinical measure of joint health and is provided by the clinician (usually a physiotherapist) at the patient's haemophilia centre so is a clinical score. In the UK it is undertaken annually and uploaded to the UK National Haemophilia Database. Details available at http://www.ukhcd.org/wp-content/uploads/2019/10/NHD_Information_Leaflet_2019_WebsiteVersion-3.pdf Separately, patients are requested to record bleeding incidents in the patient online diary "haemtrack". The two resources are combined in this analysis.  - Detailed added to methods section and now reads "Data on bleed prevalence and site were collated retrospectively from the Haemtrack patient therapy recording system and the clinical

	Haemophilia Joint Health Score from the National Haemophilia Database” p8 lines6-7
Dr. Isolde A.R. Kuijlaars, Utrecht University	
I would like to thank the authors for this paper. This paper is an observational study on joint bleeding and joint health in patients with haemophilia, with a large number of data in terms of a rare disease like haemophilia. Although the current study described joint bleeds and joint status on an individual joint level, the final conclusion of this paper is not a very new insight for haemophilia care-givers. It would be very interesting to see some data and/or some more ideas about the excluded patients who are not compliant to Haemtrack.	We are sorry but we are unable to provide further data on the less compliant cases as these data are not provided by the UK National haemophilia database. This analysis combined two datasets and while centre-reported data such as HJHS are recorded for all patients, the haemtrack records of patient reported diary entries for treatment, incidents of bleeding and “extra treatment for incidents of bleeding” are output only for those classed as “compliant with treatment delivered vs treatment recorded”. We agree that it would be helpful to compare findings for less compliant patients with those more compliant but in mitigation, one of the aims of this paper is to highlight that even in the most compliant patients, bleeding still occurs and that the ankle is significantly affected by haemarthropathy.
In addition, I am interested in the idea of the authors about annual HJHS assessment in children as the scores were very low.	Detail has now been added to the discussion  - “Similarly in children correlations between the HJHS and the Haemophilia Early Arthropathy Detection with UltraSound (HEAD-US) have shown correlations in the identification of joint pathology at the elbows and knees, however at the ankles significant difference are reported between HJHS and HEAD-US scores with underreporting of ankle joint pathology in both instances [28]. Therefore a combined approach to joint health assessment may identify pathology especially at the ankle joint prior to the progression to haemarthropathy”. P14 lines 26-27 and p15 lines 1-6. - “therefore point of care ultrasound tools such as the HEAD-US may provide early evidence of joint disease” p16 lines 12-13
Please explain 'Haemtrack compliant' in the abstract.	Haemtrack compliant: has been removed from the abstract and replaced with “Haemtrack, a patient-reported online treatment diary” p3 line 3 Haemtrack compliant is explained in the methods section

	“When CFC is administered by the patient that individual treatment is then recorded on Haemtrack, including the reason for each treatment such as prophylaxis or bleed treatment and the site of each bleed. Data recorded in Haemtrack are then integrated with NHD [22]. The 2018-2019 UKHCDO report indicated median compliance at haemophilia comprehensive care centres (CCC) and haemophilia treatment centres (HC) of 90% and 93% respectively with the NHD definition of compliance recorded use of $\geq 75\%$ of received factor concentrate [21]” p8, line 11-17
Abstract: Please clarify in more detail what this study adds to current evidence on joint bleeds and joint health and why this is interesting for the haemophilia field.	We have updated the conclusions in the abstract to highlight the relevance of the findings despite modern treatment advances, that these results highlight ongoing joint disease even in the most compliant cases and that the ankles are disproportionately affected in terms of functional impairment.
Introduction: - health related QoL = health related quality of life - page 6, lines 56/58: factor vs. Faktor	- HRQoL changed - Factor corrected
Methods: - I would suggest to describe the HJHS in more detail in de methods section	Detail added to the introduction to provide context to the methods section “The HJHS is standardised clinical assessment tool developed to assess upper and lower limb joint health status. The clinical assessments undertaken by specialist physiotherapists at 6-12 month intervals include measurement of swelling, alignment, range of motion, and muscle atrophy, and forms part of the UKHCDO haemophilia management guidelines [18, 19]. The HJHS is the most widely used score of joint health in haemophilia and has shown good to moderate correlations with radiological scores of joint disease using the Pettersson score. However haemarthrosis is not reported by the HJHS and therefore incidence of haemarthrosis and joint disease at an individual joint level are unknown [20]” page 6 lines 17-26.
Results: - Table 2 is a large table. I would recommend to include more details in the paragraph about this topic to guide the reader.	 - We have updated the text to clarify the topic for the reader and reformatted the results reducing the size of the table and incorporating AJBR data into the paragraph. P10, lines 13-20 - We agree that it would be interesting to test different subgroups but were unable to undertake sub-analyses owing to the way in which the national haemophilia database provide data. Because the UK NHD reconciles two patient identifiable data sources in-house they only provide summary data. We did not have access to individual-level data and have had to confine our analysis to the summaries provided by UKNHD. - Table 2 and 3 changed to refer only to the median

- Could it be of interest to test the differences in joint bleeds between subgroups? - HJHS: I would recommend to test the differences in HJHS scores between children vs. adults, and between joints. - Table 2 and 3: please choose for the mean or the median.	
Introduction/methods: - Please check the first use of the abbreviation HJHS	Checked and corrected on page 6 line 17/18
Discussion: - Page 14, line 18: HT? - Please add a comparison of the results to other Western populations. - What do the authors expect about the patients which are included in this study vs patients not included because they were Haemtrack non compliant?	 - HT corrected to Haemtrack in full - Comparison of results to western populations We have added a section comparison with other Western populations  - “A longitudinal evaluation of tailored frequency-escalated prophylaxis in a Canadian cohort of children aged 1.0 - 2.5 years (n=36) followed up over 10.2 years (IQR 8.5-13.6) reported median index annual haemarthrosis rates of 0.95 (0.44–1.35) which is similar to our own results. Prophylaxis treatment in Canada was driven by bleed incidence and escalated accordingly, so their treatment was more targeted and reactive [34]. The Canadian study shows that avoidance of all joint bleeding is unlikely to be possible, and in our own cohort the mean (SD) AJBR of 0.81 (1.68) and 1.00 (1.18) in haemophilia A and B children respectively, indicate that bleeding is occurring in some children even when compliant with prophylaxis. In a Dutch study of haemophiliac adults (n=62) over a 5-10 year period with a low median AJBR (IQR) 0.0 (0.0-2.0) there was still a worsening of joint health, with a HJHS increase of more than 4 points over the study period in 37.1% of patients, and with the ankle joints most often affected (30.6%) [35]” Page 16, lines 7-19 We have added a sentence describing what we might expect about the patients which are included in this study  - This study focusses, for databasing reasons, on the most compliant cases and therefore those within the broader haemophilia population likely to be suffering the fewest consequences. It might be reasonable to expect that over the twelve month study period, comparable patients who do not report or full comply with treatment may have had higher bleed rates. Consequently it would also be expected that joint health may also be worse or deteriorating at a faster rate. Compliance is important

	because it represents a gap between the availability of best treatment and impact of treatment on the consequences. Less compliant patients may require different behavioural or system-based approaches to encourage compliance and better reporting and monitoring.
--	---

VERSION 2 – REVIEW

REVIEWER	Can Ugur, Mehmet Izmir Bozyaka Training and Research Hospital, Hematology
REVIEW RETURNED	06-Sep-2021

GENERAL COMMENTS	The reviewer completed the checklist but made no further comments.
--